

# Development of an autophagy-related gene prognostic signature in lung adenocarcinoma and lung squamous cell carcinoma

Jie Zhu[1], Min Wang[2] and Daixing Hu[3]

[1] Department of Intensive Care Unit, The People's Hospital of Tongliang District, Chongqing, China
[2] Department of Respiratory and Geriatrics, Chongqing Public Health Medical Center, Chongqing, China
[3] Department of Urology, The First Affiliated Hospital of Chongqing Medical University, Chongqing, China

## ABSTRACT

**Purpose**. There is plenty of evidence showing that autophagy plays an important role in the biological process of cancer. The purpose of this study was to establish a novel autophagy-related prognostic marker for lung adenocarcinoma (LUAD) and lung squamous cell carcinoma (LUSC).

**Methods**. The mRNA microarray and clinical data in The Cancer Genome Atlas (TCGA) were analyzed by using a univariate Cox proportional regression model to select candidate autophagy-related prognostic genes. Bioinformatics analysis of gene function using the Gene Ontology (GO) and the Kyoto Encyclopedia of Genes and Genomes (KEGG) platforms was performed. A multivariate Cox proportional regression model helped to develop a prognostic signature from the pool of candidate genes. On the basis of this prognostic signature, we could divide LUAD and LUSC patients into high-risk and low-risk groups. Further survival analysis demonstrated that high-risk patients had significantly shorter disease-free survival (DFS) than low-risk patients. The signature which contains six autophagy-related genes (EIF4EBP1, TP63, BNIP3, ATIC, ERO1A and FADD) showed good performance for predicting the survival of LUAD and LUSC patients by having a better Area Under Curves (AUC) than other clinical parameters. Its efficacy was also validated by data from the Gene Expression Omnibus (GEO) database.

**Conclusion**. Collectively, the prognostic signature we proposed is a promising biomarker for monitoring the outcomes of LUAD and LUSC.

Corresponding author
Daixing Hu, hudaixing523@163.com

## INTRODUCTION

Lung cancer is a fatal malignancy worldwide and is one of the leading causes of death caused by malignant tumors. In 2019, the mortality of lung cancer still ranks first among those of all kinds of cancers in the United States (*Siegel, Miller & Jemal, 2019*). For worldwide, it is also the leading cause of cancer death among men and the second leading cause of cancer death among women (*Torre, Siegel & Jemal, 2016*). More than half (57%) of lung cancer patients

are diagnosed at the time of the distant stage (*Torre, Siegel & Jemal, 2016*). Even patients who underwent surgical resection, chemotherapy, radiotherapy and targeted therapy did not have significantly improved survival times. The five-year survival varies from 4–17%, leads to a need to explore new therapeutic targets and treatments (*Gray et al., 2019*; *Hirsch et al., 2017*). According to the histological classification, lung cancer is divided into small-cell lung cancer (SCLC) and non-small-cell lung cancer (NSCLC), the latter of which accounts for approximately 85% of all cases (*Neal, Hamilton & Rogers, 2014*). Squamous cell carcinoma and adenocarcinoma account for approximately 90% of the total NSCLC cases, which make them the most common types of lung cancer (*Neal, Hamilton & Rogers, 2014*). The poor therapeutic effect of NSCLC is mainly due to the lack of effective indicators for detecting the development of tumors at the early stage. Therapeutic progress of NSCLC is approached by the advances in the molecular field and the development of new drugs that target molecular abnormalities. But the existing treatment targets are prone to inducing resistance. New treatment markers and targets are needed to achieve better prognosis. The identification of aberrant genes has been a hot topic, in which the research on autophagy has a great prospect. Autophagy is the phagocytotic process of engulfing cytoplasmic proteins, complexes or organelles. The autophagosome, a cytoplasmic double-membrane structure, can be transported into lysosome and fusion with lysosome to generate the autolysosome (*Galluzzi et al., 2015*; *Levine & Kroemer, 2008*; *Maiuri et al., 2007b*). The degradation products can be transported back and recycled for general cell metabolism. Generally speaking, autophagy has a dual function in the apoptosis, which means it has positive and negative effects. It is suppressed by carcinogenic proteins to prevent excessive protein degradation in stressed tumor cells. Meanwhile, persistent autophagy activation leads to apoptosis (*White, 2015*; *Ye et al., 2012*). According to the present understanding, autophagy is involved in the innate and adaptive immune responses and can be induced by immune receptors such as Toll-like receptors and NLRs (nucleotide oligomerization domain-like receptors) (*Cadwell, 2016*). It takes part in the process of antigen presentation and the development of lymphocytes (*Zhong, Sanchez-Lopez & Karin, 2016*), which makes autophagy a possible target for improving immunotherapy in NSCLC. The relationship between autophagy and NSCLC has not been fully revealed, and some studies have suggested a role of autophagy in the targeted drug resistance. For example, in patients with NSCLC, EGFR tyrosine kinase inhibitors (EGFR-TKIs) and anaplastic lymphoma kinase inhibitors (ALK) can be used as effective treatments. EGFR-TKIs can induce autophagy, and high levels of autophagy after treatment with EGFR-TKIs may also lead to autophagic death of the cells (*Lee et al., 2015*). Hence, the combination of EGFR-TKIs with autophagy inducers may be beneficial. A similar synergistic effect can be observed with ALK inhibitor resistance (*Ji et al., 2014*). In this study, we revealed an autophagy-related risk signature involving six genes. This signature can be used as an independent prognostic marker for LUAD and LUSC patients. Our study indicates that autophagy may be a promising target for the treatment of NSCLC.
## MATERIAL AND METHODS

### Autophagy-related gene datasets and patient samples

The gene expression datasets and clinical information of LUAD and LUSC patients were downloaded from TCGA database on September 9, 2019. The supplementary clinical information of corresponding patients was obtained from cBioPortal (http://www.cbioportal.org). An independent microarray NSCLC cohort was extracted from the GEO database (accession number: GSE3141). Overall, the expression data from 1,102 samples (103 normal samples and 999 tumor samples) were obtained with the TCGA dataset. A total of 111 samples from the GEO dataset GSE3141 were used in the verification group. A total of 232 genes from the HADb (Human Autophagy Database) were identified as autophagy-related genes.

### Procedures and statistical analysis

A Consensus Clustering Analysis and a Principle Components Analysis were performed by the R programming language to verify the regulatory role of autophagy in LUAD and LUSC. The R package limma was used to screen the differentially expressed autophagy-related genes. Then, we carried out a series of gene functional enrichment analyses to determine the major biological attributes, including the GO and KEGG analyses. The GOplot package was employed to visualize the enrichment terms. A univariate Cox proportional hazard regression analysis was used to evaluate the association between overall survival (OS) or DFS and gene expression values. Next, a multivariate Cox proportional hazards regression analysis was performed using the candidate prognostic genes identified by the univariate regression analysis. The independent prognostic factors were determined by the multivariate Cox proportional hazards regression analysis, the regression coefficient and hazard ratios (HRs) were calculated by the Cox regression model. The prediction accuracy of the risk model was determined by time-dependent Receiver Operating Characteristic (ROC) analysis. Thus, we established an autophagy-related signature that could be a prediction model in LUAD and LUSC patients. On the basis of the signature, patients were classified into high-risk and low-risk groups according to their risk score, using the median score as a cutoff point. The relationship between OS, DFS and risk grouping was verified by the Kaplan–Meier method and log-rank test using the survival and survivalROC packages. We considered a $P < 0.05$ significant for all comparisons.

## RESULTS

### Differentially expressed autophagy-related genes

After extracting the expression values of 232 autophagy-related genes in LUAD and LUSC patients, 14 downregulated genes (DLC1, NRG3, NLRC4, DAPK2, MAP1LC3C, CCL2, HSPB8, FOS, PPP1R15A, GRID1, DRAM1, PRKCQ, DAPK1, and ITPR1) and 27 upregulated genes (ATG4D, BAK1, DDIT3, EIF4G1, IFNG, HDAC1, P4B, FADD, EGFR, VMP1, PARP1, ATC, SPHK1, BNIP3, TP73, IKBKE, PTK6, ATG9B, ERO1A, TMEM74, GAPDH, ITGB4, and EIF4EBP1) were identified. Scatter plots revealed the expression patterns of these differentially expressed genes between tumor and non-tumor tissues, as shown in Fig. 1.

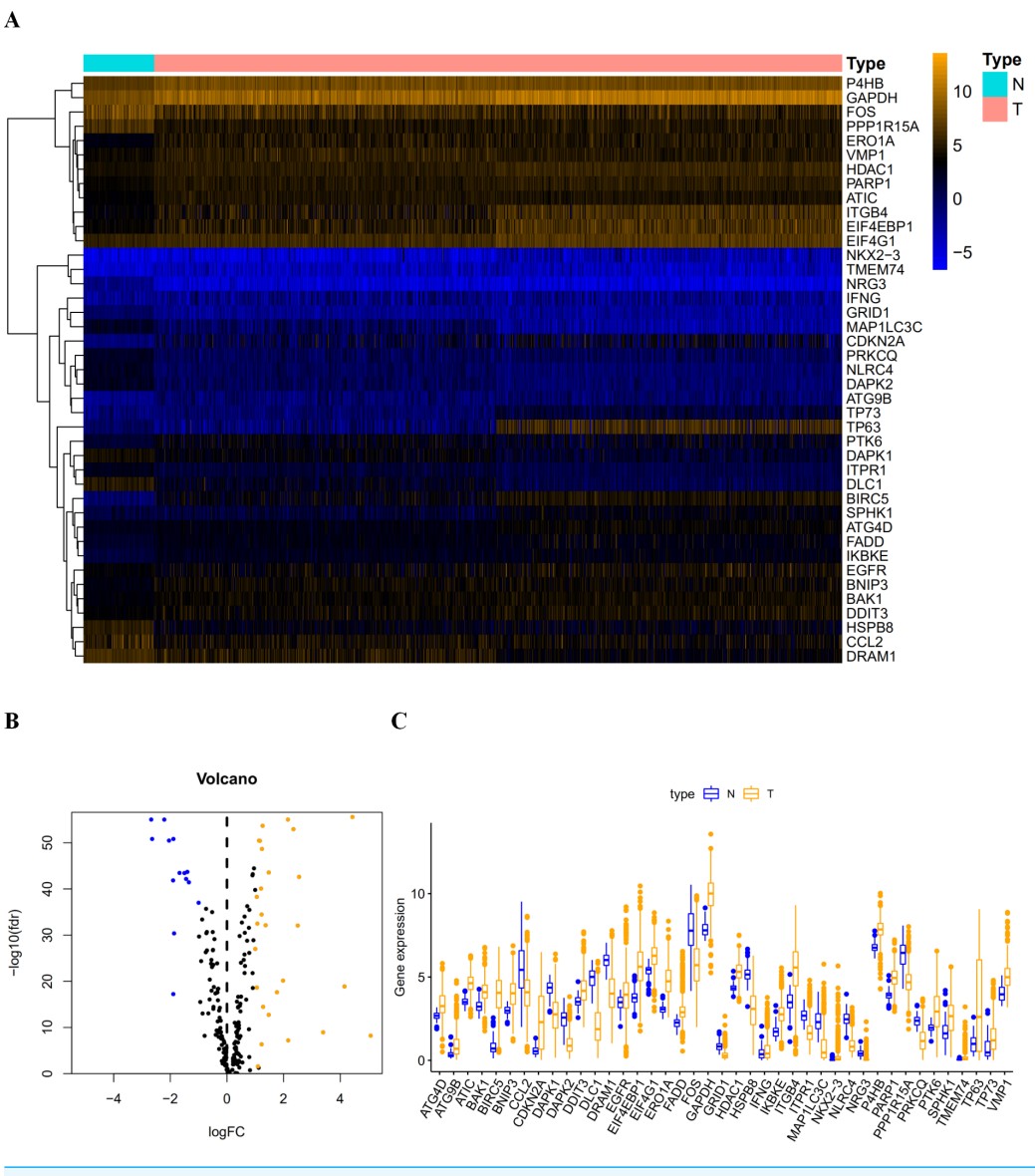

**Figure 1** **The differentially expressed autophagy-related genes.** (A) The heatmaps of these 41 differently expressed autophagy-related genes. The expression level of 41 differentially expressed autophagy-related genes was displayed. The orange color indicates the higher gene expression value while the blue color indicated the lower gene expression value. N indicates non-tumor tissues; T indicates tumor tissues. (B) The Volcano plot of the differentially expressed autophagy-related genes. The red dots indicates the high expression and the green for the low expression. (C) The boxplot of the differentially expressed autophagy-related genes. The orange color indicates the higher gene expression value and the blue color indicated the low gene expression vlaue. N indicates non-tumor tissues; T indicates tumor tissues.

## Confirmation of autophagy-related expression patterns via consensus clustering analysis and principal components analysis

By setting a K value of 2, we obtained the optimal CDF value and classified the patients into two clusters (Figs. 2A, 2B, 2C). Principal Components Analysis showed two significantly different distribution patterns. The samples of cluster 1 and cluster 2 were distributed on
the left side and the right side, respectively (Fig. 2D), suggesting that autophagy may play a role in the occurrence and development of LUAD and LUSC.

## Functional enrichment analysis of the differentially expressed genes

Functional enrichment analysis of the 41 differentially expressed genes offered a biological understanding of these genes. The GO term functional enrichment and the KEGG pathway enrichment analyses of these genes are summarized in Figs. 3 and 4.

The top enriched GO terms for biological processes were autophagy, processes utilizing autophagic mechanisms, and the intrinsic apoptotic signaling pathway. Cellular components included the autophagosome membrane, the autophagosome, and integral components of the mitochondrial outer membrane. On the basis of molecular function, genes were mostly enriched in terms of protein phosphatase binding, phosphatase binding, and p53 binding. In the KEGG pathway enrichment analysis, these genes were shown to be notably associated with pathways in the autophagy (animal), apoptosis, and bladder cancer pathways. Most of the $Z$-scores of enriched pathways were more than zero, indicating that most of the pathways were more likely to be enhanced.

## Identification of an autophagy-related risk signature for the prognosis of LUAD and LUSC

By revealing the distinct expression patterns found in LUAD and LUSC patients, we considered that identifying an autophagy-related risk signature might be useful for predicting prognosis. A univariate Cox regression analysis was performed to establish a candidate pool of autophagy-related genes (Fig. 5A). Ultimately, five genes (HDAC1, ATG4D, TP73, EIF4EBP1 and TP63) were identified as protective factors (HR < 1), while another five genes (BNIP3, DAPK1, ATIC, ERO1A and FADD) were identified as risk factors (HR > 1). Subsequently, a multivariate Cox analysis was conducted. As a result, EIF4EBP1, TP63, BNIP3, ATIC, ERO1A and FADD were identified as independent prognostic indicators for DFS and selected for development of the prognostic signature.

According to the multivariate Cox proportional hazards regression model, we obtained the expression coefficient of each independent risk gene. Our prognostic model for predicting prognosis based on the six genes was formed using the following formula: prognosis index (PI) = (−0.170 * expression level of EIF4EBP1) + (−0.057 * expression level of TP63) + (0.117 * expression level of BNIP3) + (0.170 * expression level of ATIC) + (0.214 * expression level of ERO1A) + (0.268 * expression level of FADD). We then calculated the risk score of each patient and used the median risk value as a cutoff point for classifying patients into high-risk group ($n = 732$) or low-risk group ($n = 370$), as shown in Figs. 5D and 5E. The heatmap of these six signature-related genes and the Kaplan–Meier curve depending on risk score are also displayed (Figs. 5B and 5C). A significant difference in survival between the high-risk group and the low-risk group was observed. Patients in the high-risk group had a shorter OS than patients in the low-risk group (five-year survival rate = 36.7% vs. 44.9%, $p = 0.0017$). Similar results could also be seen with the DFS (median time = 0.344 years vs. 0.512 years, $p <0.001$). The results of Kaplan–Meier analysis also showed a prognostic ability of each single gene. The downregulation of EIF4EBP1 was

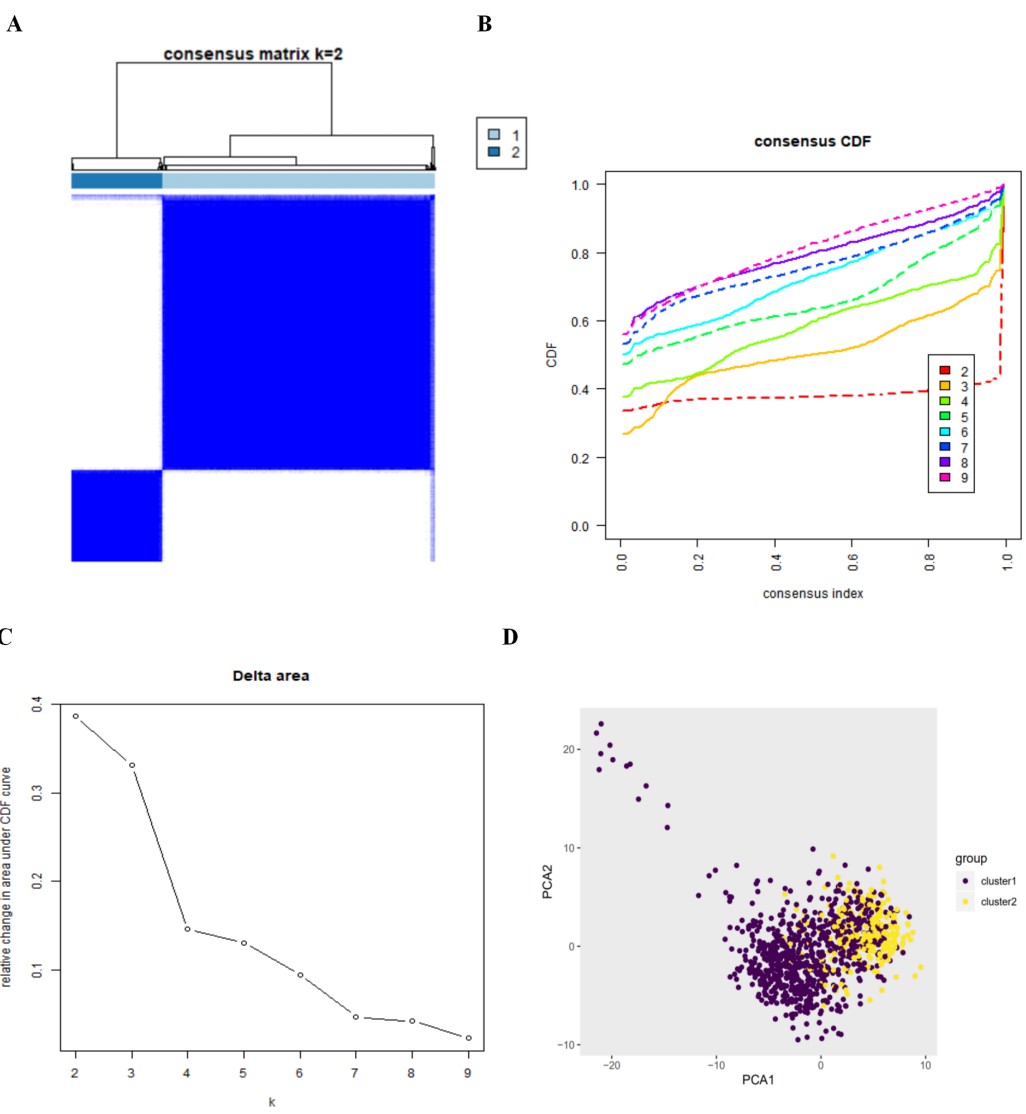

**Figure 2** **The consensus clusterin analysis and the principle components analysis.** (A) (B) (C) The consensus clusterin cnalysis of the autophagy-related genes, infering the optimal number of clusters, the lowest proportion of ambiguous clustering and the best CDF value by taking the $K$ value of 2; (D) the principle components analysis of the autophagy-related genes in LUAD and LUSC patients.

strongly correlated with inferior DFS in LUAD and LUSC patients ($P < 0.05$; Fig. 6A). Similarly, low expression of TP63 led to inferior DFS ($P < 0.05$; Fig. 6B). In contrast, the upregulation of ATIC, ERO1A and FADD indicated a decreased DFS ($P < 0.05$; Figs. 6D, 6E and 6F). However, we did not observe a significant difference in DFS with regard to BNIP3 expression ($P < 0.05$; Fig. 5C). Considering that each gene had a different value in the prognostic model, a statistical difference may not occur in each survival analysis of a single gene. ROC curves of OS and DFS were used to reveal the predictive performance of the six-gene risk signature (Fig. 7). The AUC values of the signature for OS and DFS were 0.656 and 0.671, which were obviously higher than those associated with age (AUC = 0.547
**A**

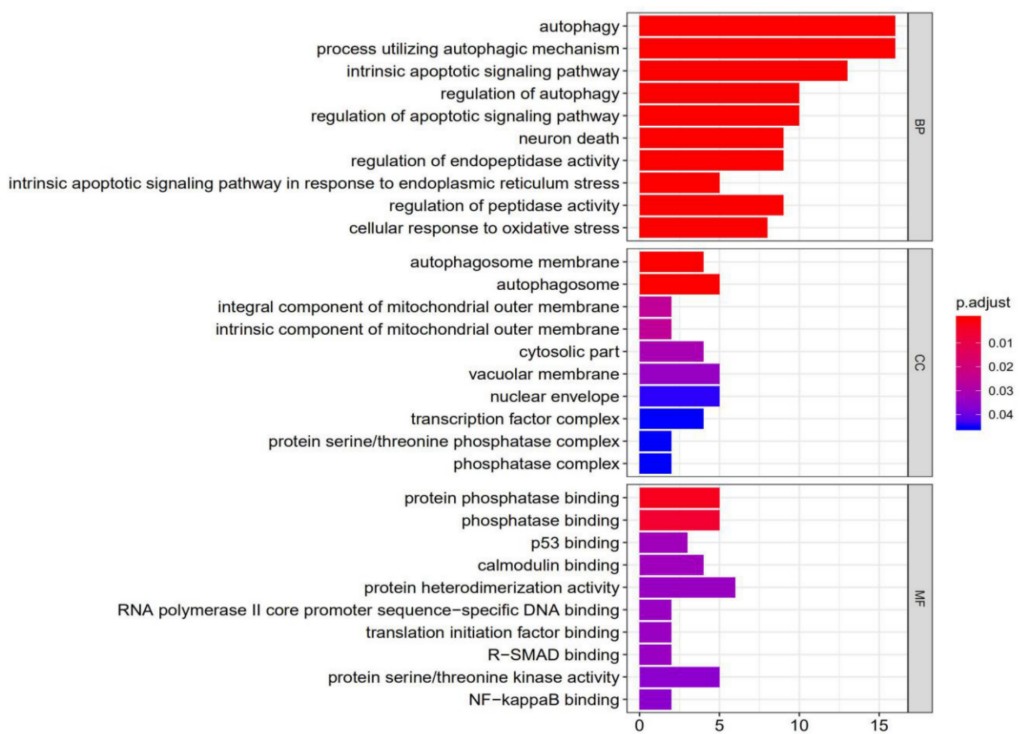

**B**

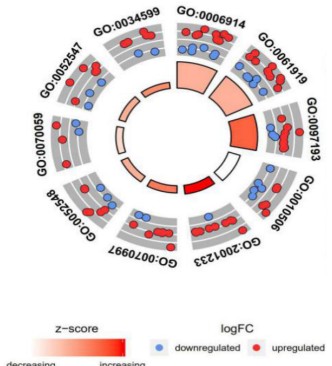

| ID | Description |
|---|---|
| GO:0006914 | autophagy |
| GO:0061919 | process utilizing autophagic mechanism |
| GO:0097193 | intrinsic apoptotic signaling pathway |
| GO:0010506 | regulation of autophagy |
| GO:2001233 | regulation of apoptotic signaling pathway |
| GO:0070997 | neuron death |
| GO:0052548 | regulation of endopeptidase activity |
| GO:0070059 | intrinsic apoptotic signaling pathway in response to endoplasmic reticu |
| GO:0052547 | regulation of peptidase activity |
| GO:0034599 | cellular response to oxidative stress |

**Figure 3** **The barplot and GO circle of functional enrichment analyses.** The barplot and GO circle of functional enrichment analyses. (A) BP indicated biological process; CC indicated cellular component; MF indicated molecular function. (B) The circle shows the scatter map of each item of the logFC of the specified gene. The red circles displays up-regulation, and the blue ones displays down-regulation. The higher the $Z$-score value indicated, the higher expression of the enriched pathway.

and 0.478, respectively), sex (AUC = 0.551 and 0.502, respectively), tumor stage (AUC = 0.634 and 0.641, respectively), tumor T stage (AUC = 0.629 and 0.648, respectively), tumor N stage (AUC = 0.578 and 0.633, respectively) and tumor M stage (AUC = 0.501
**A**

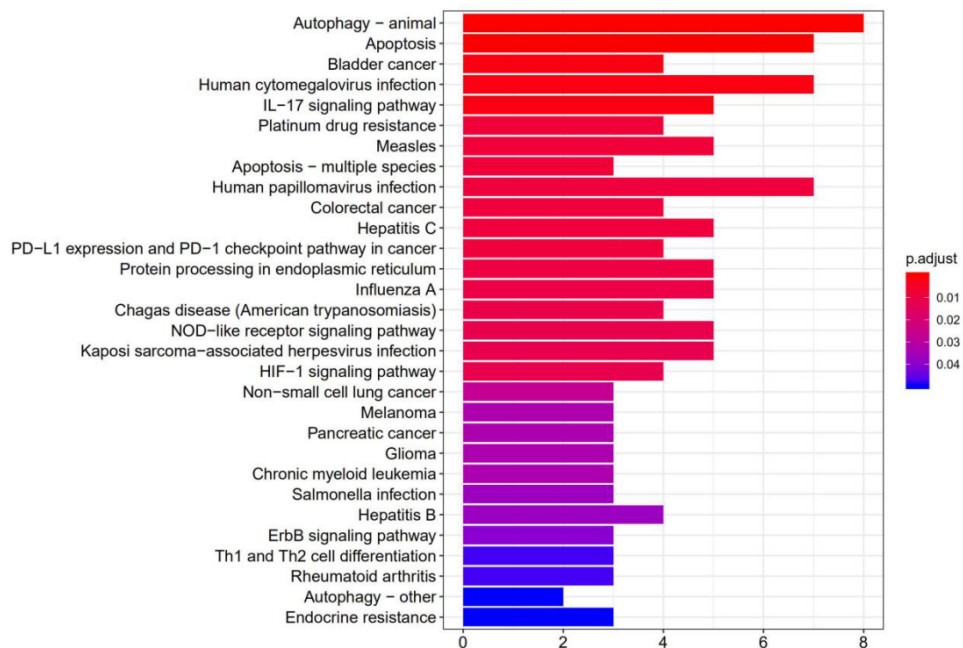

**B**

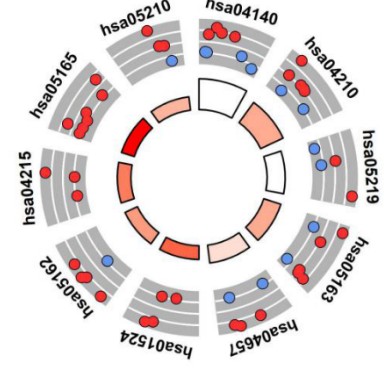

| ID | Description |
|---|---|
| hsa04140 | Autophagy – animal |
| hsa04210 | Apoptosis |
| hsa05219 | Bladder cancer |
| hsa05163 | Human cytomegalovirus infection |
| hsa04657 | IL–17 signaling pathway |
| hsa01524 | Platinum drug resistance |
| hsa05162 | Measles |
| hsa04215 | Apoptosis – multiple species |
| hsa05165 | Human papillomavirus infection |
| hsa05210 | Colorectal cancer |

**Figure 4   The barplot and KEGG circle of functional enrichment analyses.** The barplot and KEGG circle of functional enrichment analyses. (A) The KEGG analysis of differentially expressed autophagy-related genes. (B) The circle shows the scatter map of the logFC of the specified gene. The red circles display up-regulation, and the blue ones display down-regulation. The higher the $Z$-score value indicated, the higher expression of the enriched pathway.

and 0.489, respectively). These results indicated that the risk signature had a better ability to predict the survival of LUAD and LUSC patients than did clinical factors.

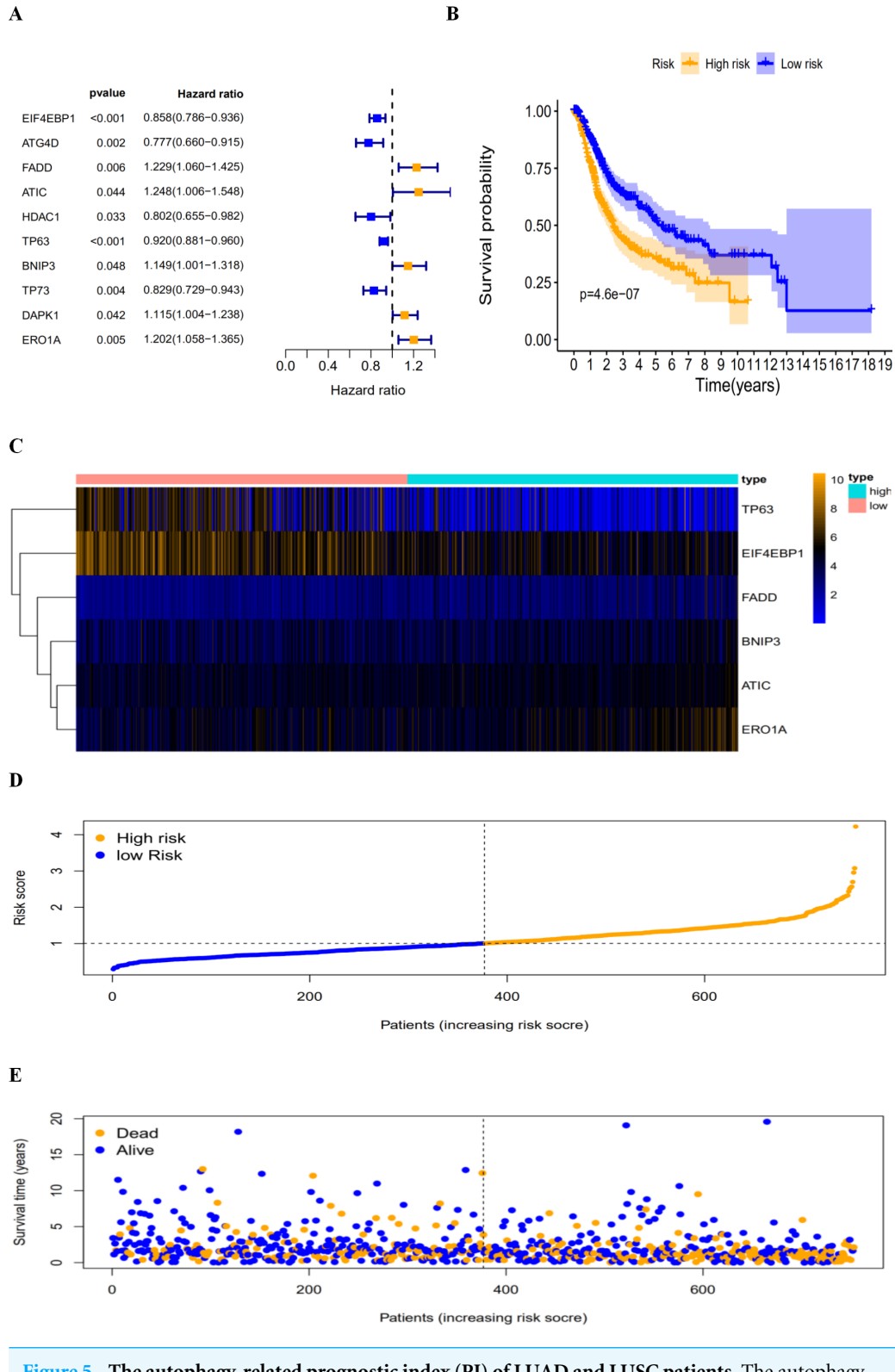

**Figure 5** **The autophagy-related prognostic index (PI) of LUAD and LUSC patients.** The autophagy-related prognostic index (PI) of LUAD and LUSC patients. (A) The univariate Cox regression analysis

**Figure 5 (...continued)**
revealed the pool of the prognosis-related genes. (B) The K-M plot represented that the high-risk group had shorter DFS than the low-risk group; (C) the heatmap of the six signature genes expression profiles; (D) the number of patients in different risk group; (E) the DFS of patients in the TCGA dataset. The orange color indicates a higher risk score and the blue color indicates a lower risk score.

## Associations between the autophagy-related risk signature and clinicopathologic features in LUAD and LUSC patients

An analysis was performed to explore the associations between clinical parameters and the risk signature (Fig. 8). The results showed that the signature was significantly associated with tumor stage ($p = 0.006$), M stage ($p = 0.004$), and survival outcome ($p < 0.001$). Additionally, Student's $t$-test analysis also indicated that these signature-related genes were differentially expressed across various clinicopathological parameters. As shown in Fig. 9, differential ATIC expression was found across different tumor stages, M stages and survival outcomes. Differential expression of BNIP3 was observed across different tumor stages and M stages. EIF4EBP1 showed different expression across different sexes and survival outcomes. ERO1A, showed differential expression across sexes, tumor stages and T stages. A difference in the expression of FADD was observed across ages and sexes. The differential expression of TP63 was related to survival outcome, sex and M stage.

## The autophagy-related signature is an independent prognostic factor for LUAD and LUSC patients

We performed a univariate Cox regression analysis and a multivariate Cox regression analysis to verify the independent predictive value of the autophagy-related signature for DFS (Figs. 10A and 10B) and OS (Figs. 10C and 10D). The univariate Cox analysis showed that the autophagy-related signature, tumor stage, and T and N stages were all correlated with the survival of LUAD and LUSC patients. Then, those factors were included in a multivariate Cox analysis, which showed the autophagy-related signature to be an independent predictive factor. Thus, our results confirmed that the autophagy-related signature could be used as an independent prognostic factor in clinical practice.

## Validation of the autophagy-related signature via an independent cohort

We calculated the risk score for each patient in the GEO dataset GSE3141 as an independent external validation using the same formula. The patients were divided into high-risk and low-risk groups based on the median risk score. The Kaplan–Meier analysis confirmed the prognostic ability of our signature once again (Fig. 11A). As expected, the high-risk patients had a lower DFS than the low-risk patients (four-year survival rate = 25.6% vs. 52.3%, $p = 0.0079$). The ROCs also showed a good ability of the signature to predict survival (Fig. 11B). Specifically, the AUC of our signature was 0.615. Because of the lack of clinical data such as sex, age, and tumor stage, we could not perform ROC analysis of other clinical factors. These validation experiments confirmed the valuable ability of our risk signature to predict the prognosis of LUAD and LUSC patients. A combined application of the risk signature and other clinical features would improve prognostic prediction of LUAD and LUSC outcomes.

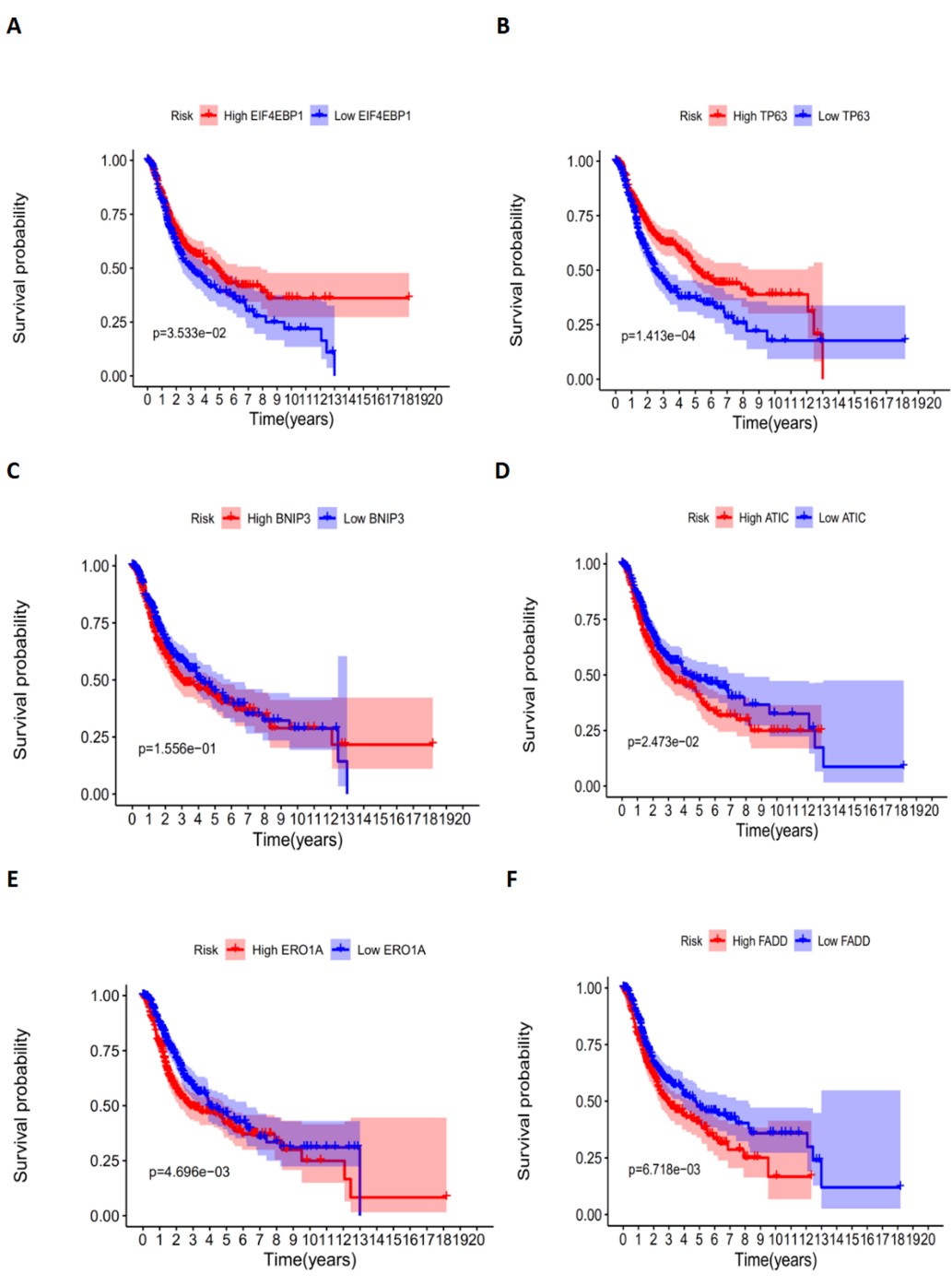

**Figure 6** **The correlation between six genes involved in the signature and DFS.** The correlation between six genes involved in the signature and DFS. The K-M plots revealed (A) the expression level and DFS of EIF4EBP1, using median separation; (B) the expression level and DFS of TP63, using median separation; (C) the expression level and DFS of BNIP3, using median separation; (D) the expression level and DFS of ATIC, using median separation; (E) the expression level and DFS of ERO1A, using median separation; (F) the expression level and DFS of FADD, using median separation.

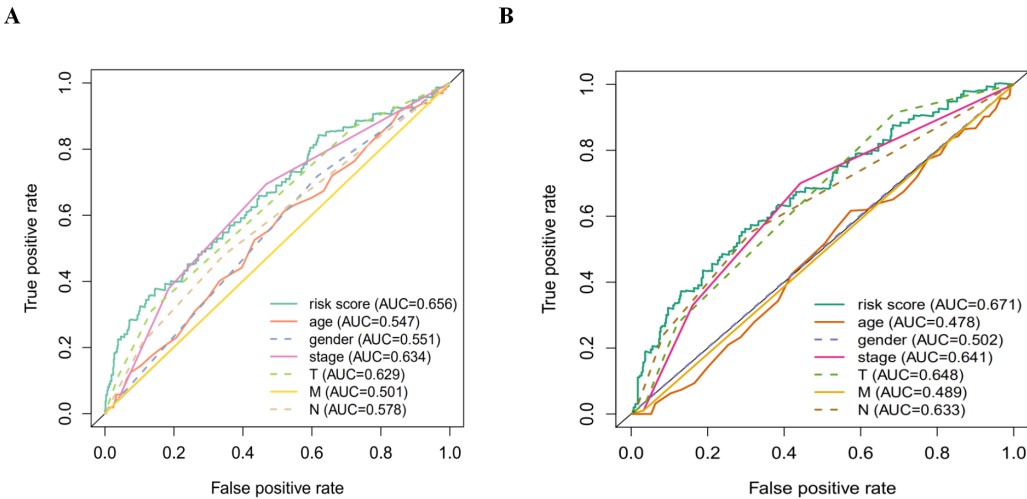

**Figure 7 The ROC analysis.** (A) The ROC analysis of OS for the signature and the clinicopathologic parameters; (B) the ROC analysis of DFS for the signature and the clinicopathologic parameters.

## DISCUSSION

Autophagy is a process carried out by cells to address nutritional deficiency and other cellular stresses. It is involved and regulated by a series of proteins and is closely correlated with a variety of cell processes and reactions. In recent years, a large number of studies have shown that autophagy is a "double-edged sword" in the occurrence and treatment of tumors. On the one hand, as a survival mechanism of tumor cells facing pressure, it plays a role in protecting cells. On the other hand, autophagy can result in killing tumor cells under certain conditions. Therefore, autophagy is considered to be a possible regulatory point for improving the therapeutic effects of tumor-targeted drugs and reducing drug resistance.

Autophagy has promise for improving the survival of NSCLC, but most studies usually focus on the role of a particular gene related to autophagy. The large-scale databases, such as TCGA and GEO, provide us with effective measures to explore gene signatures, thus providing a better understanding of the relationship between autophagy and tumors. In this study, based on the existing gene data of patients with NSCLC, we screened autophagy-related genes and identified six key prognostic genes, all of which may be possible molecular biomarkers of prognosis and potential therapeutic targets. We verified the autophagy-related genes in multiple datasets, which proved that the signature had very good prognostic ability across data from multiple centers.

The GO and KEGG analyses were also conducted to show the molecular and biological pathways enriched. The results suggested that the top enriched GO terms in terms of biological processes and cellular components were highly correlated with autophagy. On the basis of molecular function, p53 binding is closely related with the autophagy-related gene TP63 which will be discussed in detail later.

**A**

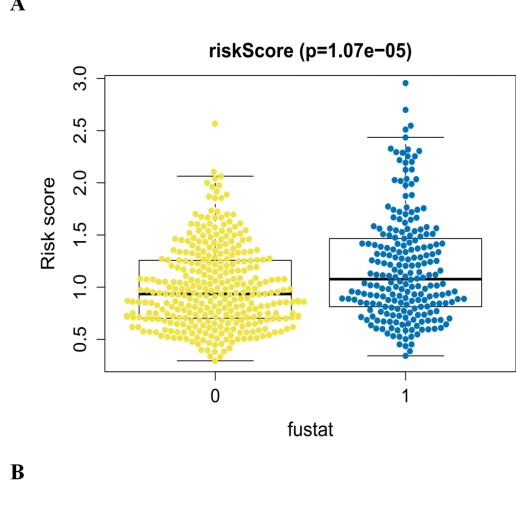

**B**

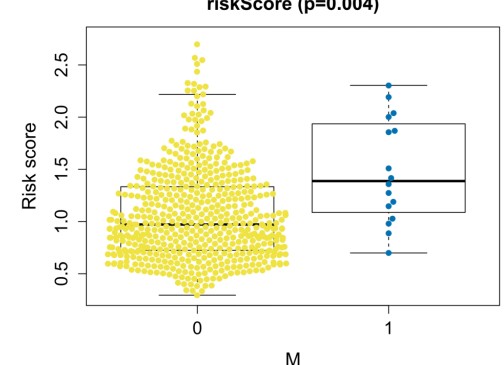

**C**

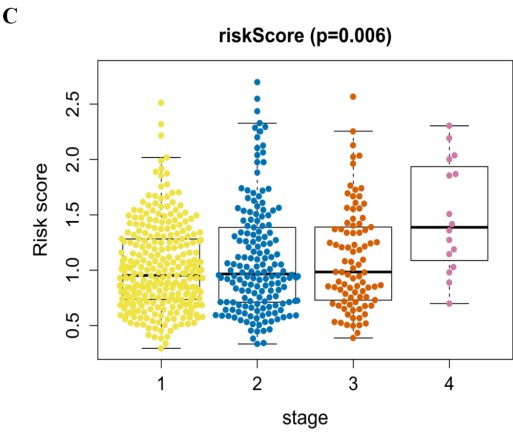

**Figure 8** **The autophagy-related signature in the cohorts.** (A) The autophagy-related signature in the cohorts stratified by survival outcome (fustat = 0 indicated alive, fustat = 1 indicated dead); (B) the autophagy-related signature in the cohorts stratified by M stages (M = 0 indicated M0, M = 1 indicated M1); (C) the autophagy-related signature in the cohorts stratified by tumor stages (1–4).

In addition, in the KEGG analysis, the most significant pathway was also enriched in autophagy processes. Because of this result, we speculated that specific autophagy pattern

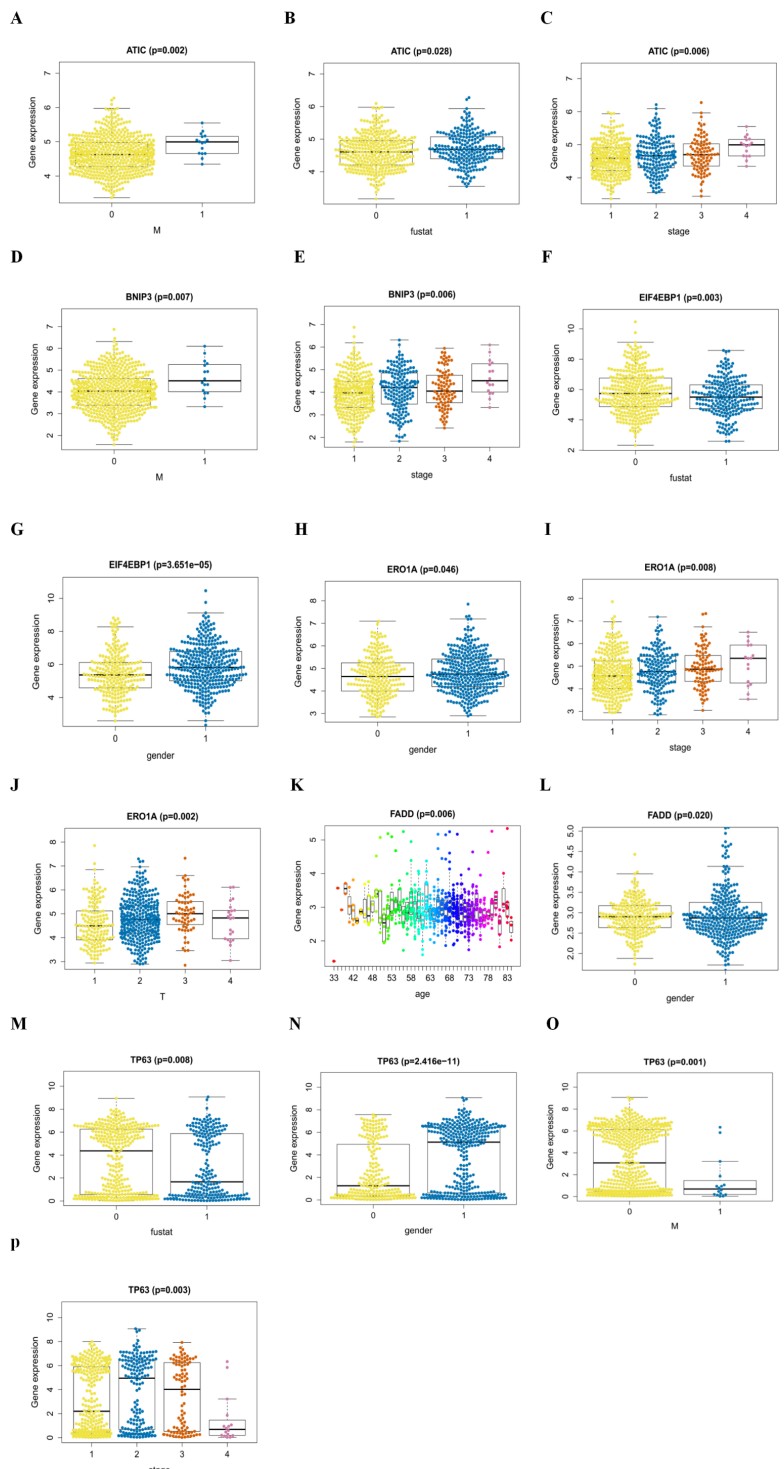

**Figure 9** **The signature-related genes in the cohorts.** (A, B, C) ATIC in the cohorts stratified by M stages (M = 0 indicated M0, M = 1 indicated M1), survival outcome (fustat = 0 (continued on next page…)

**Figure 9 (…continued)**

indicated alive, fustat = 1 indicated dead) and tumor stages(1–4); (D, E) BNIP3 in the cohorts stratified by M stages (M = 0 indicated M0, M = 1 indicated M1), tumor stages(1–4); (F, G) EIF4EBP1 in the cohorts stratified by survival outcome (fustat = 0 indicated alive, fustat = 1 indicated dead) and gender (gender = 0 indicated female, gender = 1 indicated male); (H, I, J) ERO1A in the cohorts stratified by gender (gender = 0 indicated female, gender = 1 indicated male), tumor stages(1–4) and T stages (1–4); (K, L) FADD in the cohorts stratified by age and gender (gender = 0 indicated female, gender = 1 indicated male); (M, N, O, P) TP63 in the cohorts stratified by survival outcome (fustat = 0 indicated alive, fustat = 1 indicated dead), gender (gender = 0 indicated female, gender = 1 indicated male), M stages (M = 0 indicated M0, M = 1 indicated M1) and tumor stages (1–4).

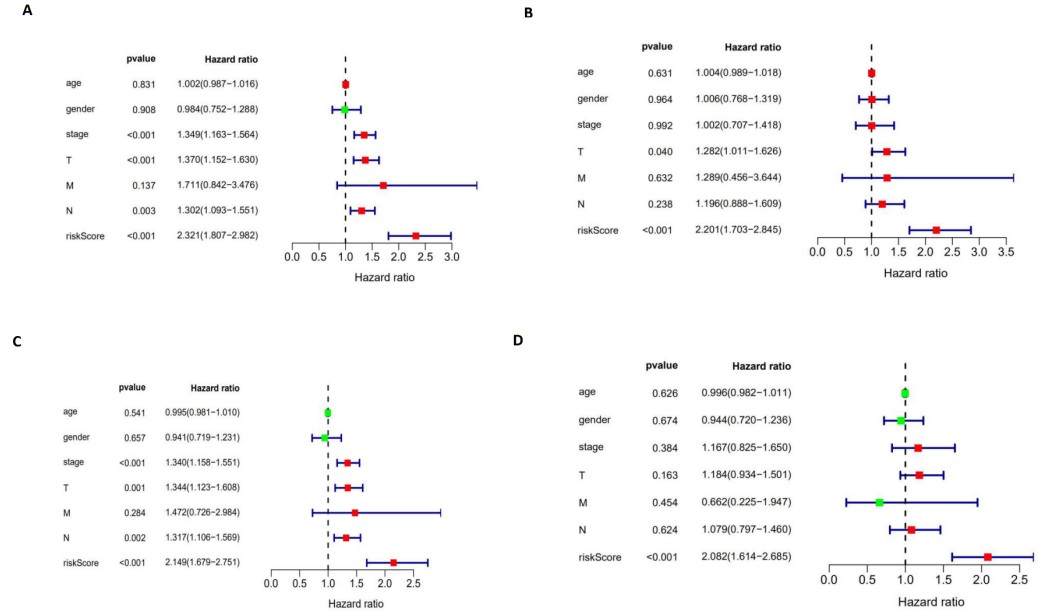

**Figure 10 The univariate and multivariate Cox regression analyses.** (A, B) The univariate and multivariate Cox regression analyses verify the independent value of the autophagy-related signature for DFS; (C, D) the univariate and multivariate Cox regression analyses verify the independent value of the autophagy-related signature for OS.

may act as tumor promoters in the occurrence and development of NSCLC. The results of the univariate survival analysis showed that ten autophagy-related genes were associated with DFS. Further multivariate survival analysis helped to identify six autophagy-related genes (EIF4EBP1, TP63, BNIP3, ATIC, ERO1A and FADD) to establish a prognostic signature, which could be used as an independent prognostic marker for NSCLC patients. However, the effects of autophagy are not immutable; they are not the same in different kinds of tumors or at different stages. Further research is still needed to explore the specific mechanism. Existing research and data reveal some roles of these related genes in autophagy or tumors.

The protein encoded by EIF4EBP1 binds to eukaryotic translation initiation factor 4e (EIF4E) and suppresses the EIF4E complex, thus affecting the mTOR (mammalian target of rapamycin) signaling pathway, which has been shown to promote tumorigenesis (*Karlsson*

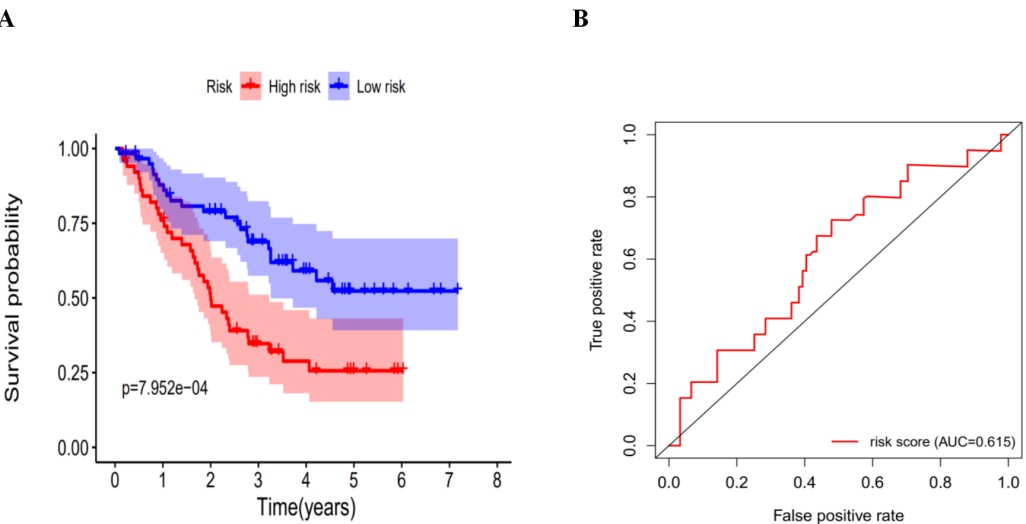

**Figure 11 The K–M curves for DFS in the high-risk and the low-risk groups.** (A) The K-M curves for DFS in the high-risk and the low-risk groups stratified by the autophagy-related signature in the GEO dataset GSE3141; (B) the ROC analysis in the GEO dataset GSE3141.

*et al., 2013*; *Karlsson et al., 2011*). EIF4EBP1 may promote or inhibit the development of tumors as a bi-functional factor (*Armengol et al., 2007*; *Cai, Ye & She, 2014*; *Martineau et al., 2013*). In general, phosphorylated EIF4EBP1 is considered to be an indicator of tumor activity, indicating a worse prognosis. Unphosphorylated EIF4EBP1 (*Cai, Ye & She, 2014*), on the other hand, is thought to inhibit tumor activity (*Martineau et al., 2013*). The TP63 gene is located on chromosome 3q27/29 and is closely related to human head, neck, esophagus, lung and skin squamous cancers (*Massion et al., 2003*; *Senoo et al., 2001*).

It encodes many subtypes of p63 transcription factors, which are members of the p53 protein family, an important hub in the transcriptional and signaling networks of the epithelial cells. Therefore, the dysregulation of TP63 is closely related to the occurrence of squamous cell carcinoma (*Romano, Solomon & Sinha, 2012*). The genome-wide analysis suggests that the genetic variant of TP63 may affect susceptibility to LUAD (*Hu et al., 2014*; *Hu et al., 2011*; *Miki et al., 2010*). However, there is still a lack of research on the specific mechanism.

BNIP3 was shown to be one of the most important players involved in autophagy. It encodes proteins belonging to the Bcl-2 family, which can regulate programmed cell death under some circumstances and may impart some pro-death activity (*Vande Velde et al., 2000*). It is linked to autophagy through three potential mechanisms. First, it can induce mitochondrial dysfunction to activate autophagy (*Scherz-Shouval & Elazar, 2011*). At the same time, through competitive binding with the BCL2 complex, BNIP3 can liberate Beclin-1 to induce autophagy (*Bellot et al., 2009*; *Maiuri et al., 2007a*). In addition, by inhibiting Rheb, an upstream activator of mammalian target of rapamycin (mTOR), BNIP3 may activate autophagy by repressing mTOR (*Li et al., 2007*). A study on early, operable NSCLC showed that the high expression of BNIP3 was an independent predictor

of poor OS (*Giatromanolaki et al., 2004*). In vitro experiments also confirmed that BNIP3 participated in lung cancer cell migration by interacting with aryl hydrocarbon receptor (AhR) (*Tsai et al., 2017*). ATIC is a protein enzyme that acts on the last two steps of the de novo purine biosynthetic pathway (*Martinez-Outschoorn et al., 2017*; *Yamaoka et al., 1997*). Little has been revealed in regard to its function in lung cancer. Recent studies have shown that inhibiting the activity of ATIC metabolites may be important for the anti-tumorigenic effects of the drug pemetrexed, which is used against NSCLC (*Racanelli et al., 2009*).

ERO1A is a main regulatory factor of protein disulfide isomerase (PDI), which is one of the most abundant proteins in the endoplasmic reticulum. Endoplasmic reticulum stress is reported to be associated with tumorigenesis in a variety of cancers, including NSCLC (*Kim et al., 2012*; *Cancer Genome Atlas Research, 2008*). As an important component of the endoplasmic reticulum, PDI is believed to be a marker of poor prognosis in patients with tumors, such as glioblastoma, breast cancer and hepatocellular carcinoma (*Shai et al., 2003*; *Thongwatchara et al., 2011*; *Xia et al., 2017*). It is not surprising that ERO1A, as the main regulator of PDI, is also associated with poor prognosis in NSCLC (*Hsu et al., 2016*).

FADD was originally described as an adapter molecule for apoptosis and is the key to transmitting death signals from cell surface receptors (*Mouasni & Tourneur, 2018*). It is closely related to autophagic cell death and tumor development. Similarly, high expression of FADD was observed in NSCLC, and it was considered to be associated with the increased invasive behavior of the tumor and a marker for predicting prognosis (*Chen et al., 2005*; *Luo et al., 2018*).

We summarized the relevant information of the six autophagy-related genes (Table 1). It can be seen that half of them have a two-side role in the development of cancer (in addition to BNIP3, FADD and ERO1A, a consistent tumor-promoting effect is present). These results are in agreement with the current consensus that autophagy plays a bi-functional role in tumors. As we mentioned previously, most of the current studies only target an individual gene. However, in view of the complex effect of autophagy, it may not be appropriate and beneficial to simply inhibit or induce some of the autophagy-related genes based on these findings. Our study suggests that autophagy-related genes may affect the cancer development through special pattern, and by which autophagy may show a consistent effect. Revealing these specific patterns can help the clinicians identify the high-risk types and use them as new therapeutic targets. Our signature based on autophagy-related genes also confirms this conjecture.

In summary, the molecular mechanisms play an important role in the relationship between autophagy and NSCLC. Our results are expected to be applied to clinical practice, which means it may suggest potential targeted autophagy therapies for NSCLC. Further investigations will provide more information of internal mechanisms. Our study first reveals that autophagy-related pattern may affect the prognosis of patients with LUAD and LUSC. And a signature is presented to help distinguish the high-risk patients. However, a limitation of this study is its retrospective nature. More prospective studies should be conducted to validate the prognostic function of autophagy-related signatures. We also encourage multi-center data to confirm our findings. More laboratory data based on the

Zhu et al. (2020), *PeerJ*, DOI 10.7717/peerj.8288

**Table 1  Introduction and summary for the six autophagy-related genes.**

| Gene | Encoding protein | Functional pathway | Function | Recent Report in cancer |
|---|---|---|---|---|
| EIF4EBP1 | a translation repressor protein binding to EIF4E | the mTOR signaling pathway | inhibit EIF4E complex and the cap-dependent translation to regulate mRNA translation | promote or inhibit the development of tumors as a bifunctional factor/ associate with poor prognosis in breast cancer (*Karlsson et al., 2011*)/ act as tumor suppressor in SCC (*Spilka et al., 2012*) |
| TP63 | the multiple isoforms of the p63 transcription factor | the metabolic pathways, like glucose metabolism, activation of TIGAR and HK II, degradation of PGM, fatty acid oxidation and mitochondrial respiration (*Maddocks & Vousden, 2011*) | activate the autophagy gene network | tumorigenesis and tumor suppression/ relate to the oncogenic potential role of SCC/ the genetic variant rs10937405 in TP63 have been found in various ethnic populations like Japanese, Korea, north Indian and British (*Wang et al., 2011*) population and to be associated with the lung cancer risk |
| BNIP3 | a proapoptotic protein belongs to the Bcl-2 family | the mitochondrial dysfunction/ the production of ROS/ the repression of mTOR | regulate programmed cell death and impart the pro-death activity/ induce autophagy | a progression marker in primary human breast cancer/ be linked with poor OS in NSCLC |
| ATIC | a cytosolic enzyme in the de novo purine biosynthetic pathway | the production of the intermediate FAICAR and IMP (*Chan et al., 2015*; *Greasley et al., 2001*) | unknown | play a significant role in the anti-tumorigenic effects in the drug of NSCLC/ be related to the poor prognosis of HCC (*Jiang et al., 2019*; *Li et al., 2017*) |
| ERO1A | a major regulator of PDI | PDI dysfunction/ unfolded protein response/ ER stress | participate in tumorigenesis | a marker of poor prognosis in some tumors, such as glioblastoma, breast cancer and hepatocellular carcinoma/ a poor prognostic factor for OS in NSCLC (*Kim et al., 2018*) |

Zhu et al. (2020), *PeerJ*, DOI 10.7717/peerj.8288

**Table 1** (*continued*)

| Gene | Encoding protein | Functional pathway | Function | Recent Report in cancer |
|------|------------------|--------------------|----------|-------------------------|
| FADD | a key adaptor protein transmits apoptotic signals | a bridge between DRs and initiator pro-caspase-8/10 (*Kischkel et al., 1995*)/ apoptosis (*Mouasni & Tourneur, 2018*)/ interaction with ATG5 (*Pyo et al., 2005*)/ a negative regulator of necroptosis (*Osborn et al., 2010*) | regulate cell cycle progression and proliferation | a cancer driver in oral, esophageal, laryngeal, and breast carcinomas (*Callegari et al., 2016*; *Chien et al., 2016*; *Prapinjumrune et al., 2010*)/ a marker for predicting prognosis in NSCLC (*Cimino et al., 2012*) |

**Notes.**

EIF4E, eukaryotic translation initiation factor 4E; mTOR, mammalian target of rapamycin; SCC, squamous cell carcinomas; TIGAR, TP53- induced glycolysis and apoptosis regulator; HK II, hexokinase II; PGM, phosphoglycerate mutase; ROS, reactive oxygen species; OS, overall survival; NSCLC, Non-small-cell lung cancer; FAICAR, formyl-5-Aminoimidazole-4-carboxa-mide-1- $\beta$-D-ribofuranoside; IMP, inositol monophosphate; HCC, hepatocellular carcinoma; PDI, protein disulfide isomerase; DRs, death receptors; ATG5, autophagy-related 5.

thought of autophagy pattern can further develop our study and provide the internal mechanisms of autophagy-related network.

### Funding
The authors received no funding for this work.

### Competing Interests
The authors declare there are no competing interests.

### Author Contributions
- Jie Zhu conceived and designed the experiments, performed the experiments, analyzed the data, contributed reagents/materials/analysis tools, prepared figures and/or tables, authored or reviewed drafts of the paper, approved the final draft.
- Min Wang performed the experiments, analyzed the data, contributed reagents/materials/analysis tools, prepared figures and/or tables, approved the final draft.
- Daixing Hu conceived and designed the experiments, analyzed the data, contributed reagents/materials/analysis tools, authored or reviewed drafts of the paper, approved the final draft.

### Data Availability
The datasets are available at GEO (GSE3141) and at the TCGA.

Our retrieval strategy for our TCGA dataset is as follows: in the Cases section, we chose bronchus and lung as Primary Site, TCGA as Program, TCGA-LUAD and TCGA-LUSC as Project, adenomas and adenocarcinomas squamous cell neoplasms as Disease Type. In the Files section, we chose transcriptome profiling as Data Category, Gene Expression Quantification as Data Type, RNA-Seq as Experimental Strategy, HTSeq - FPKM as Workflow Type.

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
