# Peer review of "Development of an autophagy-related gene prognostic signature in lung adenocarcinoma and lung squamous cell carcinoma"

_PeerJ, doi:10.7717/peerj.8288_

## Round 0.1 · original submission · Minor Revisions

Please address critiques of all reviewers and revise manuscript accordingly

Reviewer 1 ·

Basic reporting

no comment

Experimental design

no comment

Validity of the findings

no comment

Additional comments

The manuscript Zhu et al provides a comprehensive analysis of autophagy-related gene expression profiles in two types of lung cancer and their prognostic value in predicting the clinical outcome. The authors run a comprehensive bioinformatic analysis and identify 6, autophagy-related genes and then classify them based on their positive or negative prognostic contributions to both diseases. They then establish that this signature is independent and highly predictive. The manuscript is nicely written, and the data are well-presented. Only a few minor edits are required before its accepted.

- Minor spelling and grammar check are required. Many factual sentences in the Introduction and Discussion are mentioned without appropriate referencing. The authors are encouraged to run a more careful revision of the writing.
- Although the authors do a great job on discussing the functional role of the 6 autophagy-related genes in the Discussion, they are invited to arrange those genes in a table, identifying their contribution to the autophagy pathways, and recent evidence on their prognostic roles individually. Some genes, e.g., FADD is not directly a part of canonical autophagy pathways, and while the authors mention that, the table will be an easier way to look at the role of each gene. This will be of great help to the reader.
- Have the authors thought about how their analysis reflects on the current debate on the contribution of autophagy to cancer outcomes? Whether the induction/inhibition of autophagy would have a better outcome? How is this prognostic tool will be interpreted in the context of the different functional roles of autophagy in response to cancer therapy?
- The authors are encouraged to include more on their future studies and the major limitations of their prognostic tool.

Reviewer 2 ·

Basic reporting

In this paper, the researchers analyzed clinical data using a univariate Cox proportional regression model to identify autophagy-related prognostic genes for lung adenocarcinoma and lung squamous cell carcinoma. Overall the paper has provided enough data analysis to support the conclusion, however, there are a few things needed to be addressed to increase the quality of the paper.

Experimental design

See below

Validity of the findings

See below

Additional comments

1. The language needs to be examined thoroughly by a native English speaker to avoid gramma mistakes.
2. The author needs to be careful with giving full descriptions of abbreviations, for example, what is AUC in lane 37?
3. It will be better if the author can list the identified genes in the abstract part
4. The author needs to provide more recent information in the introduction, for example, in Lane 44, the statistical data was from 2012, which is really out of date

Reviewer 3 ·

Basic reporting

no comments

Experimental design

Can the authors describe more extensively how the risk score was calculated ?

Validity of the findings

The validity of the findings can be confirmed by applying the signatures in the clinical treatments / does the authors have any plans to proceed in this direction?

Additional comments

can the authors add future work to the paper?

---

## Round 0.2 · accepted · Accept

Since all critique were addressed and the manuscript was appropriately amended, I am glad to inform you that the current version of the manuscript is acceptable.